# The Evaluation Gap in Astronomy—Explained through a Rational Choice Framework

Julia Heuritsch

Research Group "Reflexive Metrics", Institut für Sozialwissenschaften, Humboldt Universität zu Berlin, Universitätsstraße 3b, 10117 Berlin, Germany; julia.heuritsch@hu-berlin.de or julia.heuritsch@gmail.com

**Abstract:** The concept of evaluation gaps captures potential discrepancies between what researchers value about their research, in particular research quality, and what metrics measure. The existence of evaluation gaps can give rise to questions about the relationship between intrinsic and extrinsic motivations to perform research, i.e., how field-specific notions of quality compete with notions captured via evaluation metrics, and consequently how researchers manage the balancing act between intrinsic values and requirements of evaluation procedures. This study analyses the evaluation gap from a rational choice point of view for the case of observational astronomers, based on a literature review and 19 semi-structured interviews with international astronomers. On the basis of the institutional norms and capital at play in academic astronomy, I shed light on the workings of the balancing act and its consequences on research quality in astronomy. I find that astronomers experience an anomie: they want to follow their intrinsic motivation to pursue science in order to push knowledge forward, while at the same time following their extrinsic motivation to comply with institutional norms. The balancing act is the art of serving performance indicators in order to stay in academia, while at the same time compromising research quality as little as possible. Gaming strategies shall give the appearance of compliance, while institutionalised means to achieve a good bibliometric record are used in innovative ways, such as salami slicing or going for easy publications. This leads to an overall decrease in research quality.

**Keywords:** Reflexive Metrics; evaluation gap; anomie; research behaviour; research quality

## 1. Introduction

*Reflexive Metrics* (cf. [1,2]), a subfield of science studies, acknowledges that one needs to consider that indicators, such as publication and citation rates, do not merely describe, but also prescribe behaviour. In other words, numbers are performative (e.g., [3–5]). That is why indicator use in performance evaluations can affect research behaviour, knowledge production processes and even research quality. Two accounts for these performative effects are relevant for this study: the evaluation gap and constitutive effects. The evaluation gap [6] refers to a potential discrepancy between what researchers value as scientific quality as opposed to what is being measured by indicators. Such a discrepancy may lead to so-called unintended consequences, such as questionable research practices, involving "goal displacement" and "gaming" (e.g., [7,8]). Constitutive effects [9] describe the shaping role of indicators. For example, by defining what scientific quality is, indicators may shape practitioners' perception of quality research.

In order to study the effects of indicator use in the field of astronomy, in an earlier publication [2], I performed a case study of international astronomers at Leiden Observatory and explored the intrinsic and extrinsic motivations to perform research. This study found evidence for an evaluation gap, since what astronomers value as quality research diverges from what indicators measure. Astronomers are driven by intrinsic values such as "curiosity, truth-finding and 'pushing knowledge forward'" [2] (p. 176) and what indicators count gives an extrinsic motivation to perform research. Because their intrinsic values remain

as their "ideals", but they must serve performance indicators in order to stay in academia, astronomers "hold two opposing notions of science: the 'ideal' one which corresponds to their intrinsic values and the 'system' notion." (ibid.: 177). Consequently, a third notion arises—the balancing act. Astronomers try to manage a balancing act between fulfilling the requirements of the evaluation system, on the one hand, and not compromising their internal notions of what good quality research means to them, on the other hand.

Because Reflexive Metrics is a relatively new field in the sociology of science, there have been a handful of studies on the efficacy of indicators in the field of astronomy, but not yet a comprehensive study on what effects indicator use has on research quality in astronomy. In a study [10] that follows up on Heuritsch [2], I reviewed such papers, which, for example, study the publication activity of astronomers [11] or the indicators' efficacy in predicting an astronomer's scholarly performance [12]. The set of literature that this review is missing concerns the *moral economy* of astronomy which is studied by McCray [13], Atkinson-Grosjean and Fairley [14] and Baneke [15].

The term moral economy draws back to Thompson [16], for which a moral economy is related with a mentalité—"the expectations and traditions—that structured and mediated interactions between the consumers and producers of life's basic needs" [13] (p. 686). This includes what rights people have and how (non-)economic relations are regulated through social norms. Kohler [17] adapted that concept to the activities of experimental scientists. Based on [16,17], Atkinson-Grosjean and Fairley [14] (p. 148) write that a moral economy is a "system of shared values, traditions, and conventions about ways of doing, being, knowing, and exchange" held by a moral community. While the authors define "community" as something held together by "group values rather than overarching social structures or institutions" (ibid.: 148), Baneke [15] (p. 3) states that the moral economy of a scientific community "includes scientific, institutional and [...] moral values". Atkinson-Grosjean and Fairley [14] (p. 150) distance themselves from a material determinism, where "material conditions dictate the course of events, but rather that they demarcate the bounds of the possible". Moral economies are shaped by material, societal and cultural constraints and conditions—contrary to Merton's [18] view, where the scientific institution is "distinctly separate from the wider social environment and operates within internally established universal norms" [14] (p. 149). One of the functions of a moral economy is to "provide provisional maps for navigating the messy, contingent spaces where societal and scientific values are negotiated" (ibid.: 169). As the authors point out, a moral economy in science may be sensitive to external conditions, such as the "grants-based structure of 'normal' academic inquiry" (ibid.: 149). This raises questions about "research ethics, scientific authority, unintended consequences, power differentials, and cost-benefit ratios" (ibid.: 149). According to McCray [13], in astronomy required resources to conduct scientific work include large sums of money to fund the construction of telescopes. In the USA these were, historically, built by private institutes or universities instead of the state, resulting in privileged access to those who were affiliated to the respective facility. Only after the National Science Foundation (NSF) started to provide federally funded observatories in the mid-1950s, access to telescopes was made available to a larger population of astronomers. While this was a first step to democratise the telescope time allocation in astronomy, the distinction between those who have privileged access to telescopes based on their institutional affiliations (the "Haves") and those who have to compete for telescope time (the "Have-Nots") resulted into two different moral economies in astronomy [13,14].

This study builds upon the research conducted by Heuritsch [2] and is a follow-up analysis of the interviews conducted by Heuritsch [10]. It expands on the effects that indicator use has on research quality in astronomy by employing a rational choice framework. Rational Choice Theory (RCT) is a sociological theory of action, which explains how social phenomena on a relative macro level emerge out of social phenomena from a relative micro level and how, in turn, they can influence micro phenomena [19].

RCT explains the role of institutional norms in shaping behaviour: Through the *orientation function* of an institution the actor knows what is right or wrong in a situation

and how to behave [20]. Institutions provide *scripts* for certain situations, including what *mode of action* to choose. Most of the times, choosing the mode is not a conscious decision, but prescribed by the norms present in the *logic of the situation*. Only in situations where it is worth it to spend cognitive resources, for example when there is no script available to the actor, the actor goes into a reflected mode. This may also happen, when the actor disagrees with the norms or the prescribed scripts. In addition to the orientation function, institutional norms also have a function that endows the situation with meaning; the mere idea of the existence of a legitimate orderliness defines the meaning of behaviour in the specific cultural frame [20]. Because meaning is introduced by institutions, meaning is socially constructed and by legitimating the institution, it functions as the *internal anchor* of an institution. Those actors who subjectively agree with the institution, its norms and what it represents, support the *internal anchor*. Actors who do not agree with that specific institution do so, because they were socialised differently or decided to follow different, often contradicting, norms. For those actors, an institution needs an *external anchor* in the form of enforcement mechanisms like sanctions, in order to make them follow the rules. However, when the absolute value of the gain from deviating from the norms is higher than that of the loss, actors will choose not to follow the norms. In other words, astronomers will deviate from the mode of normative behaviour, when that deviation has a higher utility.

I use RCT as a framework, because it provides the appropriate tools to reconstruct the conditions for the evaluation gap in astronomy, observed in my earlier study [2]. In the follow-up [10], I subsequently set the basis for this study by analysing the *logic of the situation* of a researcher in astronomy, which is the first step in finding a sociological explanandum for the evaluation gap. This paper performs step two and three: deriving behaviour from the logic of the situation and explaining how individual actions aggregate to a collective phenomenon. Building up on the research of the relevant conditions under which astronomer perform research [10], this study analyses how these structural conditions in academical astronomy shape research behaviour and impact research quality.

Therefore, this study's **research question** states: "How do the structural conditions prevailing in academical astronomy shape research behaviour, in particular with respect to a focus on research quality versus quantity, in such a way that an evaluation gap arises on aggregation level"?

With RCT, I aim to provide a more integrated explanation of the evaluation gap in astronomy. I analyse what effects indicators use in research evaluation has on research quality in astronomy, by shedding light on the balancing act astronomers undergo in their daily research life between staying true to their intrinsic values and serving performance indicators. I will also discuss what all this means for the moral economy in astronomy. Understanding the causal mechanisms that lead to the evaluation gap, with this and future studies I may be able to inform policy makers about the structural conditions that need to be taken into account and fostered—in practice—in order to support astronomers to perform research that is of good quality and integrity.

A few terms and concepts used in the result section may be unfamiliar to the reader and are therefore introduced at this point:

**The theory of Anomie**

Esser [20] refers to the theory of anomie [21] as a useful concept to explain how actors adapt or deviate from the present structural conditions (i.e., the three components of the logic of the situation). "Anomie" occurs when there is a dissociation between how to legitimately meet personal or cultural goals and (possibly contradicting) institutional norms. This may be either because it is not clear what institutionalised means (e.g., capital) [22] [1] are necessary to reach the goal or because the capital is limited in a way that (some) people simply do not have enough capital to meet the requirements set by the institutional norms. In either case, *anomie* weakens the internal anchor of institutional norms and deviant behaviour becomes more attractive. Merton [21] defines five behavioural patterns as a response to anomie, which differ in their degree of normative and deviant behaviour: Conformity, Innovation, Ritualism, Retreat and Rebellion.

**Gaming**

The fact that the need to be accountable can lead to gaming is not new (cf. [7,8]); however, I frame the "indicator game" [1] in terms of the RCT framework. The fact that resources are limited and that control over them is assigned according to the rules of the institution, including what indicators count, leads to a *material bond* [23] between astronomers. Figure 1 illustrates such a material bond by depicting a situation where two actors have a certain interest and at the same time exercise as certain control on two resources. The presence of such a material bond implies that actors need to employ strategies in order to maintain or gain control over resources. The scarcer the resources, the more efficient need to be the strategies. Any kind of strategic behaviour can be analysed using *game theory* [23]. Thereby strategies are rules about actions, available for a player in a certain situation in a specific game. Therefore, strategies are not actions, but roadmaps that determine actions as re-actions towards the actions of another player.

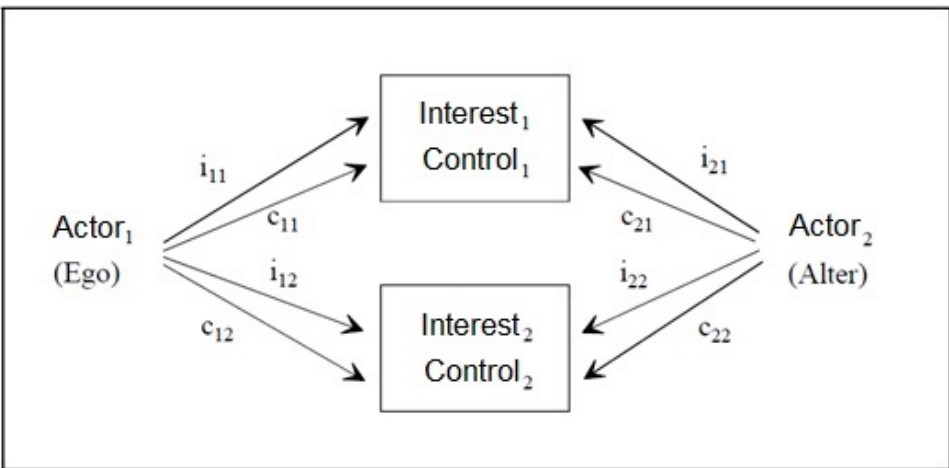

**Figure 1.** "The system of a social situation" (created by the author, based on [23]).

**Cognitive Dissonance**

Cognitive dissonance is a psychological concept that is closely related to the sociological concept of anomie. While an anomie is a dissociation between the cultural goals and the institutionalised means, a cognitive dissonance may occur when an actor experiences any clash of pieces of information or values—whether normative or not. There are three ways to resolve cognitive dissonance: denial of the incoming information that contradicts available knowledge; the re-valuation of one preference in favour for the opposing one; finding arguments for why pursuing one of the preferences will in the long-term also benefit the pursuit of the other. All of those are aspects of rationalising one's situation, which is to find a rational justification for one's behaviour through highlighting some of the arguments for the behaviour at the cost of the arguments against the behaviour. As Esser [19] points out, what alternative one chooses is an act like any other that happens according to the *expected utility theory*.

This paper will be structured as follows: First, I give a brief recap of the methods used. A detailed description can be found in the prior study [10]. This is followed by the results from the interviews and the literature review. Weaving previous studies together with this one, the results are framed according to RCT. This style was chosen consciously in order to give a comprehensive and integrated rational choice picture of academic astronomy. Section 3.1 describes the second step in finding a sociological explanandum for the evaluation gap in astronomy by deriving research behaviour from the astronomer's situation. Section 3.2 is dedicated to the third step—the *logic of aggregation*—explaining how indicator use in astronomy, through shaping astronomers' research behaviour, leads to an evaluation

gap, where research quality is sacrificed for quantity. The final section contains a discussion and conclusion.

## 2. Materials and Methods

This study is a follow-up in the line of research by Heuritsch [2], who conducted nine interviews with astronomers employed by the Leiden Observatory between December 2016 and February 2017 and is based on the data collected by Heuritsch [10]. The latter data set consists of 19 semi-structured interviews with international, observational astronomers, conducted between June 2017 and December 2018, and described in detail in the methods section by Heuritsch [10]. These included face-to-face interviews at several international conferences; in Prague in June 2017, in Vienna in August 2018 and in Bremen in October 2018. At the conferences, a letter of invitation was provided to the organisers who distributed it via the conference-apps and/or news updates. In addition, the chairs of sessions which are related to observational astronomy were asked to pass on the invitation text, which resulted in interviews via Skype between October and December 2018. When selecting interviewees from respondents, I paid attention to ensuring a variety in gender, seniority and country of employment (see Appendix A: Table A1). Empirical findings from the interviews were compared and complemented by a comprehensive review of the astronomy-related science studies literature.

Interviews were held in English and amount to 50–100 min in length, were transcribed by a company and coded in MaxQDA according to Mayring's qualitative content analysis [24], involving circles of inductive and deductive code development. Deductive category application was employed on basis of the codes used by Heuritsch [2] and were further developed during analysis. Thereby some codes were dropped, some added and some specified. The final set of codes can be found in the Appendix B (Table A2). Additionally, two researchers from my research group intercoded the interviews to increase the reliability of the analysis.

Accounting for the fact that observational astronomers need observational data to do their work, interview questions address the challenges astronomers face when acquiring those data. Other questions concern themes such as evaluation procedures, which indicators are important therein, what obstacles astronomers face when doing their research, gaming strategies and what motivational factors drive them in doing their research. The complete list of interview questions can be found in the Appendix C (Table A3). Since I employed semi-structured interviews, not necessarily all questions were answered by all interviews, when for example it was apparent that the question would not be relevant for the interviewee. However, most interviewees at least gave a brief answer to each question. Answers referring to the structural conditions prevailing in astronomy were analysed by Heuritsch [10], while answers referring to research behaviour and consequences thereof, are analysed in this study. The MaxQDA codes thereby helped in structuring and guiding this analysis, by shedding light on relevant aspects of research behaviour and consequences.

## 3. Results

The *logic of the astronomer's situation* [10], sets the basis for an astronomer's research behaviour. Before I continue exercising step two (Section 3.1) and three (Section 3.2) of finding a sociological explanandum for the evaluation gap in astronomy, I provide a recap on the first step—the logic of the astronomer's situation.

To summarise the author's [10] findings, the author studied the internal drivers, material opportunities, norms and cultural frame. With respect to the internal constituent of the logic of the situation, their findings agree with Heuritsch's [2]: astronomers are driven by their curiosity about uncovering the laws of the universe. Given that intrinsic drive and their realist attitude, they define research quality as fulfilling *three criteria*: 1. Originality of the research question, 2. Correctness of the performed study, 3. Good communication of the performed study. All those criteria ensure that knowledge gets pushed forward, which is one of the main intrinsic drivers of an astronomer. With respect

to material opportunities, Heuritsch [10] finds that there are various forms of capital that astronomers depend on in order to perform good research. First, astronomers need economic capital, such as funding, time, access to telescopes, data and technical equipment such as computers, strong internet connection and software packages. Second, they further need cultural capital in the form of expert knowledge, programming and data analysis skills and (tacit) knowledge about institutional norms. Third, astronomers also need a good research team (including supervisor, PhDs and postdocs) and collaborators, which bring more access to more capital. Fourth, they need symbolic capital in order to climb up the career ladder and gain other forms of capital such as telescope time. Symbolic capital is based on recognition and reputation as reflected by various performance indicators. Metrics which count in astronomy are publications and authorship, citation rates and impact, allocated telescope time which count the same as grants and rewards. Finally, astronomers need luck, because luck increases the chances for success and the more successful, the more capital an astronomer receives due to the Matthew effect [25], which can be summarised as the "rich get richer" phenomenon. With respect to institutional norms, Heuritsch [10] finds that the communication and publication infrastructure is relatively well organised. However, in practice the infrastructure or time is often lacking to share adjunct data or reduction code one used for analysis. On top of that, not all studies are publishable. In fact, negative results can often not be published, which may lead to reinventing the wheel. In order to receive telescope time to acquire data, an astronomer needs to show a good track record, be affiliated with a university that has privileged access or collaborate with an astronomer who does. The author concludes with identifying three tension relationships an astronomer may be subject to, and which arise from the fact that capital is limited, and that the publication infrastructure dictates what is publishable and what is not. Those tensions result from the cognitive dissonances between: competition versus collaboration, guaranteeing usefulness versus risky and innovative ideas, pressure versus freedom and finally, primary versus secondary code (truth versus recognition; [26]), which may lead to the tension between focussing on quantity rather than on quality. How astronomers respond to these tensions will be discussed in Section "Dealing with the Tensions".

### 3.1. Astronomer's Research Behaviour

The *logic of the astronomer's situation* [10], sets the basis for an action. By setting the rules of the game and endowing behaviour with meaning, institutions structure expectations, interests and evaluations of actors. They direct behaviour by defining what is good or bad behaviour, what is right or wrong. The action theory I use is the *expected utility theory* [19], where the action that brings the highest utility will be chosen. In order to explain what effects indicator use in research evaluation has on research quality in astronomy we must first study their research behaviour. This section will delineate what behavioural contexts the logic of the situation shape, what motivations astronomers de facto follow in performing research, what conflicts they experience and how they respond to them.

I start this section by showing how the external components of the *logic of the situation* give an extrinsic motivation to perform research, in addition to the intrinsic motivation astronomers have due to their intrinsic drive to perform research out of curiosity (cf. [10]). As Davoust and Schmadel [11] (p. 11) point out, "there may be other motivations for publishing a paper, in addition to that of informing the community of new scientific results or ideas". Taubert [26] distinguished between two motivational factors for an astronomer to assume the author role: 1. The extrinsic motive of acquiring symbolic capital through securing priority [27]. 2. the intrinsic motive of disseminating new knowledge others can build on (quality criterion 3; [10]). Taubert [26] relates the extrinsic motive with the secondary code, recognition, and the intrinsic motive with the primary code, truth. Note that praise has been found to be positively correlated with intrinsic motivation as well, as when we are recognized for our work, our work appears more fun [28]. However, once recognition is needed as symbolic capital in order to move one's career forward, it becomes an extrinsic motivational factor as well (cf. [10]).

> "Because it seems the only thing that is important is number of publications and so you have to do these if you want to continue." [Int-PhD1]

The author discusses that the need to publish leads to publication pressure and Heuritsch [2] points out that this gives an extrinsic motivation to performing research. When the survival of an academic career depends on recognition through publications and citation rates ("publish or perish"), the secondary code recognition may only be defined through those indicators. In the case where indicators constitute what counts as recognition, a tension relationship between primary and secondary code may arise (cf. [10]). This can be then translated into a tension between intrinsic and extrinsic motivation. Cronin [29] (p. 559) writes that Darwin and "countless other scientists, both then and now, have experienced [the recurrent tension between intrinsic and extrinsic motivation] in the course of their careers". The discussion of how internal and external factors drive astronomers' research behaviour is the aim of this section.

### 3.1.1. Anomie

When looking at the five suggested patterns to cope with anomie to examine which could be a useful concept to describe the research behaviour of astronomers, we can easily dismiss *ritualism* and rebellion. Due to astronomers' intrinsic motivation to perform science, there is little grounds for them to comply with norms, simply for the norms' sake. Early career researchers generally are not in the position to question the institutional norms in the sense of being able to discuss them with older colleagues in a meaningful way [e.g., Int-PhD1, Int-PhD2, Int-PhD3 and Int-Postdoc1] or to not comply with them if they want to survive in academia. Int-Faculty4 also admits that his supervisor was "too important for [his career] to use as a sparring partner, [so he] couldn't really argue with him" whether it was about content or norms. Established research may get away with non-compliance every now and then. To put it in Int-Faculty3's words, he keeps "those norms as broad as possible" and is willing to accept the consequences:

> "So, I always look at data, to me that's the key, on the other hand, frequently I find that my interpretation of data maybe different than other people but I [Coughing] try to give an honest interpretation but I always look for different ways of interpreting it. *In science just like in culture there are norms. And I'm quite comfortable in keeping those norms as broad as possible.*"

However, in general I do not observe any serious attempts of *rebellion*, since there is not much chance to rebel against performance indicators and surviving in academia at the same time. Which leads us to *retreat*. Retreat is in fact common in science—many astronomers (consider to) leave. However, because I only interviewed those astronomers who remained in science, I could at most capture those who consider to leave academia. That is why our focus will not be on *retreat*. Now we are left with *conformity* and *innovation*. Let us revisit the meaning of normative behaviour for an astronomer before we can determine if any of those two patterns fit (better).

As a general approximation, normative behaviour in astronomy includes sticking to Mertonian norms and committing to the cultural goals of conducting impactful science. One should write a minimum of one innovative paper a year, which peers thoroughly check for errors and legitimacy. Further, following a "golden child trajectory" [2] (p. 166), where one career step follows another smoothly, and having been at prestigious universities (e.g., [2,30]) looks good on paper. When using other people's research that was published in papers (unlike data and software), one has to cite it. Using other's research happens to build on their results in order to push knowledge forward. Collaborators are "friendly competitors" [Int-Faculty8] who support each other in that endeavour by complementing each other's capital. However, as pointed out at the beginning of this section, there may be other motivations to publish a paper than pushing knowledge forward and having societal impact:

> "We are in a society that we have to publish our results from different reasons. Yeah, to be visible as a scientist but also to share your science." [Int-Postdoc1]

Because quality is difficult to measure, academia has adopted indicators that shall serve as proxies. Section 3.2 will demonstrate that the discrepancy between what astronomers would define as quality and what is measured by indicators leads to an evaluation gap (cf. [2]). The need to serve those indicators gives an extrinsic motivation to publishing papers, citing colleagues and acquiring other forms of relevant capital, which may have negative effects on research quality. Hence, astronomers find themselves in an anomie; they want to follow their intrinsic motivation to push knowledge forward, while at the same time following their extrinsic motivation to comply with institutional norms. That is why Heuritsch [2] concluded that astronomers need to perform a *balancing act* between both motivational drivers in order to survive in academia, while at the same time not compromising their values. I dedicate the next subsection on analysing the balancing act further. This much can be revealed already: it seems like astronomers use institutionalised means in *innovative* ways in order to keep up the appearance of *conformity*.

### 3.1.2. The Balancing Act

The following sub-sections elaborate on how astronomers manage the balancing act between serving quantitative indicators and producing good quality research and how they respond to the tensions arising from the misalignment of astronomer's quality criteria and what indicators measure.

**The Indicator Game**

In order to perform the balancing act between the intrinsic motivation to push knowledge forward and the extrinsic motivation to serve indicators to have a career in academia, I observe astronomers to deploy various strategies, which is also called *gaming*.

Here, gaming means to serve the indicators for the sake of increase in reputation (the secondary code) and survival in academia. Thereby, the roadmaps to play this indicator game need to keep up the appearance of *conformity*, but to stay competitive, institutionalised means are used in *innovative ways*. I will now elaborate on such strategies that I observed.

Astronomers may not only be in competition for resources with each other, but also with other sciences. They need to convince politicians to spend money on a scientific field, whose aim is not produce any economic outputs [15]. That is why astronomy depends on outreach and popularization of research findings to gain popular support and on adroit lobbying to secure funding. As a consequence, outreach and lobbying have been well-developed in tandem and support each other. Roy and Mountain [31] (p. 20) write that "astronomers are salespersons when they push for a given project and deploy strategies to make their project the best in the field". The authors thereby identify four motivational factors astronomers draw on to attract and justify funding for their big science projects: the quest for knowledge, the quest for achievement, the quest for survival, and the quest for power. The *quest for knowledge* is linked to the astronomers' curiosity and arguments related to this motivational factor draw on "our natural curiosity and a general desire to understand the world around us" (ibid.: 14). The quest for achievement is related to the astronomers' intrinsic drive to push knowledge forward: "A second motivation for funding research is to satisfy the need to achieve something big—to do something because it has not been done before, or doing it ten times better than before" (ibid.: 16). As Int-Faculty10 points out, pushing knowledge forward may help humanity to survive and to improve the quality of life. When those arguments are not enough, astronomers push their case with arguments concerning the *quest of survival*. They point at examples for spin offs and the "relevance of their work to protect the Earth from the most threatening danger that can face our planet over long time scales" (ibid.: 17).

> "But astronomy really helps the day life.", "We have to invent the day by day devices that are more efficient, that have …, that they use less power." [Int-Faculty10]

Lastly, the *quest for power* is used as a political driver: astronomers "can invoke national pride to show that their communities and respective countries are among the best performers in the world, or are in need of new facilities to maintain their leadership" (ibid.: 18). National astronomical communities and lobbies have developed the ability to formulate consensus priorities that are attractive to politicians and funding agencies. Roy and Mountain [31] (p. 18) point out that "in this game, astronomy is at an advantage because [...] the public holds a very favourable view of the discipline". Heidler [32] (p. 24) writes: "Astronomy draws its support from the high media and public interest, which is based on its ability to produce suggestive pictures and address philosophical issues".

Int-Faculty10 explains that the "politically correct" answer to the question on how to secure funding is to demonstrate the usefulness of the scientific project to the taxpayer in terms of being "transformational". This argument draws on all four quests. However, Int-Faculty10′s "politically incorrect" answer is that one needs to "lobby", especially if one needs funding for large instruments. Drawing on the four quests is not enough if resources are scarce. Having friends among politicians helps to convince them why the funding would be a good investment.

> "And usually the benefit is advance in science. You know politicians understand very easy issues, how to build a road to go from Milan to Rome. But if you say I build a telescope to understand the black hole they do not really understand very well." [Int-Faculty10]

Heuritsch [10] finds that instrument builders are at a disadvantage in receiving funding and citations as compared to those astronomers who merely observe. In order to keep up with the citations of those who merely conduct science and do not build, a scientific target needs to be in mind when the telescope is built. Referring to the target also helps in selling the telescope to funders. Other strategic considerations regarding funding concern its efficient use. Int-Faculty8, for example, explains that in New Zealand one postdoc is as expensive as five PhD students, which is why the university invests in favour of the PhDs and the postdocs need to find their own funding.

I also observe gaming strategies when it comes to promotions and collaborations. Int-Faculty2 points out that he "made a mistake" when he did not search for other job offers in order to be promoted in his current position: "[It] doesn't matter how good you are. You can be very good if you don't have an offer from somewhere else, it's a much higher hurdle [to get promoted]". While the head of his institute said that he isn't "playing dirty enough", he refused to play along:

> "I always refused playing any dirty games in contrast to a lot of other people I know and never pushed anyone else down. I raised everyone who worked with me up."

Int-Faculty2 observed "pushing others down" in collaborations; sometimes leaders of collaborations, who are in a more established position than PhD students, put the blame on students when something goes wrong and at the same time praise themselves with the successes coming out of the collaboration. Another strategy to get promoted is to work together with prestigious people:

> "Because now what is happening, [...] people are trying to game the system. So for example I know of people who work only to *please four or five prominent people in the field*. Who are their collaborators. So they just work like there's—they are faculty members but they work like postdocs of those guys. [...] Because they know that when the promotion case comes up they will the-they are going to be asked to write letters of recommendation and they're going to say very nice things about you because you've just worked to please them. So I've not done that and I think I have suffered in my career due to delay in my promotions." [Int-Faculty7]

Int-Faculty12 adds that students also decide for projects, depending on what is "more beneficial" for them, which may include working for more renowned scientists. The same goes for choosing collaboration partners. As explored by Heuritsch [10], collaborators are important to obtain access to resources, enhance one's research portfolio and to publish more papers than one could alone. Int-Faculty2 reports that he does not even perceive publication pressure, since publications are "just coming in" due to his collaborators. Those may not be first author papers, but they nevertheless boost one's publication rate. The negotiation of authorship is also strategic; while the lead author is listed at the top and others who contributed listed thereafter, people who did not write any text, but who made the collaboration, observation or funding possible, are also among the authors [Int-Faculty8].

Publications are one of the biggest grounds an astronomer can play the indicator game on. As Int-Faculty8 points out, the name of the game typically goes as follows: "Publish, publish a lot of papers and get, you know, a huge number of papers under my belt". One strategy to get there is to choose the research topic, such that one can get an "easy publication" [Int-Faculty1] out of it. This includes avoiding risky topics in favour of sexy ones. Int-Postdoc2 reports that flashy topics get attention, especially if published in *Nature* or *Science*. Another strategy involves publishing immature results in order to get papers out more quickly (cf. [2]). Int-PhD1, Int-Faculty1 and Int-Faculty2 report that it is normal to publish results even if one is not totally confident about them:

> "Yes, for what I know everyone has to publish, the more, the better position in the short list. (Yes, I see okay.) And that's one aspect that I don't like from researchers. [. . . ] because we are maybe it's preferred to publish something that maybe you are not so very sure, or you are not so very confident about what you are saying but it seems that the most important thing is to publish not to do research." [Int-PhD1]

Yet another strategy to get more papers published is salami slicing (e.g., [33]). Int-PhD1, Int-PhD3, Int-Faculty1, Int-Faculty2, Int-Faculty5 and Int-Faculty12 report to have cut up their papers into smaller portions. However, astronomers do not only do that in order to get more papers out of their research (which is the definition of salami slicing), but also to increase the communication value (3rd quality criterion) of the publications [2]. Int-Faculty2, for example, explains that brief and to-the-point publications are better readable and comprehendible than longer ones. Int-Faculty5 and Int-Faculty12 agree with Int-Faculty2's attitude by adding that often it is better to get the research out so that the community can give feedback and can build upon that knowledge. Int-Faculty12 points out, however, that often it is hard to draw the line where to cut up the research and where to wait for more data from the telescope to put all the results together in one paper. Because the "barrier to actually going ahead and publishing something is not very high" [Int-Faculty12], Int-Faculty6 complains that there are too many papers containing little scientific value. Int-PhD3 and Int-Faculty1 admit that they committed salami slicing. Int-PhD3 explains that his supervisor wanted him to do that too, so that Int-PhD3 and his associated Master student could both be first authors and that the latter would be able to apply for a fellowship. Int-Faculty1 explains that he was not proud of salami slicing, but that "there is no sainthood in being a scientist". Int-PhD2 observed others doing salami slicing and she finds it "ridiculous", so she would not want to do that herself. This supports Heuritsch's [2] conclusion that astronomers approve the publication of immature or cut-up results when this increases the communication value of a publication, in terms of getting valuable feedback and being able to improve the paper or getting the results quicker to the community so that others can build on the published results. Salami slicing, where publications are cut up solely for the sake of increasing one's publication rate is not approved when that decreases the quality of the paper.

Heuritsch [2] found that there are three time-frames that set the boundary conditions for the publication process: 1. the race for priority, 2. the time-frame of a PhD's or postdoc's temporary contract and 3. a telescope application deadline. Int-Faculty12 explains that one always needs to find a "sweet spot" between results to be "sufficiently interesting" and

the "schedule pressure on the student side". Since results can always be improved, one needs a "judgement call" about such a decision, which is "driven by the job-cycle": "So there's certain times of the year and certain times in a student's career where they need to be more time sensitive about getting the results out". Int-Faculty7, Int-Faculty8 and Int-Faculty10 also emphasise that it is important to make sure that PhDs get enough first author publications during their temporary contract.

An astronomer may not only want to boost their publication rate, but also their citation rate. Int-Faculty6 explains that "the best paper to write is one that has an error in it that's not completely obvious, because then everyone will catch it and will feel obliged to point it out and they have to cite you". Int-Postdoc1 complains that a referee may reject a paper, when the referee works on a similar subject, and does not want the author to claim priority.

All of the above are examples for gaming strategies that astronomers employ in order to enhance their career chances. They do so out of the extrinsic motivation to accomplish what bibliometric indicators measure, even though they perceive them as poor measures of quality [10], in order to stay in academia to perform research. Performing research out of curiosity drives their intrinsic motivation. The balancing act then is the art of playing the indicator game, while at the same time compromising research quality as little as possible. The balancing act comes in many facets. One facet is that the means may justify the ends. Int-Faculty1 and Int-Faculty8, for example, explain that, especially at the beginning of the career, one may had to perform assigned and not freely chosen research, to then find collaborators that do research that is interesting to you. You may also serve the indicators for a while until you are established enough to be freer in your research.

> "Sometimes you have to be opportunistic. Sometimes you have to go for easy publications that you can claim that you are successful, you are good, then you'll get funding. And then you will finally have time to do what you want to do." [Int-Faculty1]

> "So with this comfort buffer [i.e., having many collaborators], the publications come out anyway. They tick all those boxes for the non-important people. So I can focus on what I want, what I think is important." [Int-Faculty2]

Int-Faculty2 emphasises that he "prepared" for the interview by looking up his bibliometric record. He did that, despite asserting how "stupid" all these metrics are, which shows that he is aware that importance is attributed to such records nevertheless. Int-Faculty3 agrees that a good publication record is relevant, that "certain compromises" are necessary, but that one does not need to obey to the evaluation system completely.

> "I wouldn't conclude that you have to in order to succeed, succumb to such things as publish-or-perish. It's relevant but it's just as relevant you know, do you like to drive, well you still have to obey the speed limit that maybe a nuisance or you know stopping in a red light even though no one's in the intersection I mean there's certain compromises." [Int-Faculty3]

**Dealing with the Tensions**

An important aspect of this study is what these "compromises" (i.e., the balancing act) entail and what impact they have on research quality. Heuritsch [10] discusses several tension relationships that may result from the *logic of the situation*: 1. Collaboration versus competition; 2. Guaranteeing usefulness versus risky projects; 3. Primary versus secondary code. Part of the balancing act of an astronomer is to find ways to deal with those tensions. Often, when humans find themselves stuck between opposing (whether intrinsic or extrinsic) values, expectations or contradicting pieces of information, they tend to harmonise those knowledge- and value systems. In such a case one struggles with the so-called *cognitive dissonance* (cf. [19,20]).

In the case of the astronomer, cognitive dissonances arise due to diverging intrinsic values from what bibliometric indicators measure, which leads to the above-mentioned tension relationships. There is an anomie (see Section 3.1.1), due to the dissociation between

how to meet the three quality criteria, while at the same time serving the bibliometric indicators in a competitive way.

In addition to the three tension relationships, I would like to point out two more cognitive dissonances. First, it seems that, despite the publication pressure most interviewees feel quite free in their research. Int-PhD1, Int-PhD2, Int-Postdoc2 and Int-Faculty4 report that during their PhD they feel/felt free to experiment, learn and to pursue their own ideas. At the same time, interviewees mentioned that they were free "as long as publications come out" [Int-Faculty2]. Int-Faculty1 makes a point in not treating his PhDs as "slaves" and "broadcasting a free environment", but putting very much emphasis on "getting into the publishing business" because "this publish-or-perish thing is a real thing":

> "And to be successful they have to have more. I mean, fulfilling the minimum requirement never helps you getting into the real business. You have to be an over-performer, to be honest. So, that's my advice to my students." [Int-Faculty1]

Int-PhD3 reports that he feels like "living from pressurizing moment to pressurizing moment". He feels restricted in his research, because he does not feel like he has the time for quality research:

> "It is paper based, so it is quantity based and not quality based.", "Then you have to do it quickly in that way, like everybody does and that is the problem."

The quality of papers gives rise to the second cognitive dissonance I would like to point out here. During the interviewees, papers were mentioned as "important" or "impactful" when they received many citations or published in a journal with a high journal impact factor (JIF). However, when asked what their highest quality paper is, many interviewees mentioned that they would nominate one that did not necessarily receive the highest amount of citations or JIF (e.g., Int-Postdoc1, Int-Postdoc3, Int-Faculty5, Int-Faculty8 and Int-Faculty12), but one that instead had impact on the public or when something really had an impact in the sense of pushing knowledge forward.

> "And then so even though [...] it doesn't have a particularly high citation at all it doesn't matter. I reckon-, I look at that and go, that was something which, you know, I came up with myself. And I had that, I had that, that, that fire within to, to pursue it. And that certainly happened about two or three other times in the papers where you've got that real sense of, no, this is something new and exciting and I'm learning stuff. And it's not another run-of-the mill paper, which is good and publishable material, but it's just another incremental advance." [Int-Faculty8]

> "I mean, I personally know that many researchers do not consider their most cited paper to be their best paper. I myself fall in the same loop. My best paper doesn't have the largest number of citations. And funding agencies don't care about lowly-cited papers. So, there is no clear answer how much, do they support this kind of high-quality research." [Int-Faculty1]

Int-PhD2 reports that she was on a selection committee for a professorship, where the committee members affirmed that it is important to look at the quality of papers, but at the end of the day, they always came back to their person's citation and publication rates.

> "So yeah they were always like saying, yeah okay we're not going to focus on publications only because it's not a qualitative but only quantitative, that's not the most important thing but in the end it was." [Int-PhD2]

I conclude that what counts according to performance indicators is so incorporated that in the astronomer's mind there are two conflicting definitions of quality present: one where quality is defined through their three quality criteria and one where quality is equated with what indicators measure. Furthermore, astronomers may downplay the feeling of publication pressure as something inherent to the system, to be able to feel less restricted in their research process.

Let us come back to the balancing act. As pointed out above, gaming strategies shall give the appearance of *compliance*, while institutionalised means how to achieve a good bibliometric record are used in *innovative* ways. They best fit the third way of how to resolve a cognitive dissonance: by favouring the pursuit of the secondary code (reputation) in the short-term, one hopes to get into a position where one can focus on the primary code in the long-term. The balancing act of an astronomer then can be described as the act of choosing how much of the value "primary code" can be compromised in favour of the value "secondary code". When facing choices between "high risk, high gain" versus "guaranteed publication", "sharing data" versus "securing priority", "increasing the quality of the paper" versus "publishing as quick as possible", an astronomer will decide according to the *expected utility theory*; how much compromising of the primary code is it worth it order to have a gain in the secondary code.

### 3.2. Resulting Collective Phenomenon

The last step of the sociological explanation of how new macro-phenomena come into existence is to formulate *transformation rules*, which describe how the actions of many individual actors constitute a *collective phenomenon*. In this case—academical astronomy—institutional norms, build the basis for the transformation rules, since they structure research behaviour. It is the very purpose of institutional norms to be followed without questioning in order to guarantee smooth interactions between actors and for the actor to reduce the cognitive effort decision making would otherwise cost. Through institutional norms collective phenomena come into being, simply because they prescribe certain social processes that follow from an individual's *situation*. The transformation rule then amounts to assuming regularity of those empirical social processes, which are treated as a logical consequence from institutional norms [19].

Indeed, I observed that astronomers generally comply with institutional norms. They find it "completely normal" [Int-Faculty6] to be assessed by quantitative indicators (cf. [10]) and indeed, especially early career researchers are not in the position to discuss evaluation procedures with older colleagues (see Section 3.1). Given that astronomers are realists [2], who "like to stress their status as passive observers: they only gather information that can be received from the universe" [34] (p. 11), it does not come naturally for them to reflect upon the performativity of indicators. Those who do, would not know how to have a better evaluation system (e.g., Int-PhD2, Int-PhD3 and Int-Postdoc3) or give up in pondering about that, since they feel like they cannot change anything.

"And so even if you, fix it [i.e., introducing software citation] by encouraging citation counting, they haven't fixed the attitudes of the people who are doing the evaluation. The people who are gatekeeping." [Int-Journal; astronomer currently working for a journal]

Rather, I observed that astronomers accept the institutional norms, such as being evaluated by performance indicators, as part of "the system" [2]. Indicators need to be served to climb up the career ladder, in order to do research.

"Okay. I mean, in the publishing process mostly it's fine. You get used to what, you, you get used to what's expected." [Int-Faculty12]

"I'm sure that [sacrificing quality for quantity] happens a lot. Which is a bad thing as we all know, but since this is the world we are living in and this is still the system, although they were always saying like no we're not going to look to number of papers, but they always did in the end so. Yeah we're far from alternative ways of choosing people so." [Int-PhD2]

However, because of the discrepancy between what astronomers define as quality research, and what indicators measure, astronomers need to perform a balancing act between serving the indicators and producing quality research at the same time. As I discussed in Section 3.1, this includes various gaming strategies and results in short-term favouring the secondary code at the cost the primary code. This may have various consequences, such

as a decrease in research quality, biased funding decisions and stress, as elaborated in the following sub-sections. This section will end by demonstrating how individual short-term favouring of the secondary code leads to the evaluation gap in astronomy.

3.2.1. Decreasing Research Quality

Astronomers affirm that they would do their best not to compromise research quality, despite the publication pressure. Int-Faculty7 reports that he never faced a situation that diverged from his notion of quality. He was always in control and lucky enough that he worked with professors for whom the number of publications was not too important. With his students, he wants them to take full responsibility, but only checks for obvious mistakes. He also does not push the students into a certain direction—he communicates "the ethics of the problem [that quality is important], not so much the astronomical skill". Int-Faculty2 reports he only published when he is happy with the results, even when his collaborators push for a publication earlier. Int-PhD3 explains that he did not publish a paper in his first PhD year, since he likes being thorough and precise and eventually got his paper accepted "as is". Int-Faculty6 asserts that the standard way of doing science is to generate double-checked and reproducible research and that if people in his collaboration were to compromise quality, he would rather take his name off the paper.

> "But I want more than an apparent high quality paper. I want high quality research, high quality data.", "Whereas I know at the back of my mind, for this data point I had to do something inconsistently with something other and okay even though the paper would appear high quality-. That should not be the goal, right? The goal should be that our research is high quality." [Int-PhD3]

> "The best quality I can. It's not high quality but the best as I can. Okay everything is right [i.e., there are no errors]." [Int-PhD1]

> "So, I fight hard to have quality over quantity as much as possible, but I often fail because of the pressure." [Int-Faculty1]

Int-PhD2 also reports that she would only want to publish, once that she is happy with the quality of the paper, but that she feels pressured by her supervisor to publish quicker. Hence, despite the good intentions, I do observe a decrease in research quality due to the need to fulfil quantitative goals. As pointed out by Heuritsch [10], while methodological standards and the publication infrastructure, with its reviewing system and publication of replicable results, shall support the orientation towards the primary code of truth finding, the evaluation of astronomers through quantitative indicators leads to an orientation towards the secondary code of recognition. As Taubert [26] points out, advancing one's career opportunities (through the secondary code) is a goal that could motivate the publication of research of lower quality. The "competition [for resources] is bound to have a strong effect on the publishing activity of astronomers" [11] (p.11).

Int-PhD2 and Int-Postdoc1 report that truly bad papers would not get accepted and that handing in such a paper would give a bad impression (and hence a bad impact on reputation). But if it is "okay" [Int-Postdoc1], it usually does get accepted, even if it contains some mistakes due to the "rush" [Int-PhD1]. To ensure priority as early as possible a higher risk of containing errors is accepted for publication in ArXiv [26]. Int-Faculty2 believes that "the publication system completely fails sort of all the time", because the majority of papers shouldn't get accepted—his rejection rate as a reviewer is 2–3 times larger than usual. Next to the potential of containing errors, Int-PhD1 and Int-Postdoc1 explain that there are many articles, that are not clear in their message (having a negative impact on the 3rd quality criterion).

The focus on sexy topics may be another reason for a decrease in quality. Flashy topics attract more attention, especially if published in journals, such as *Nature* or *Science*, which have a higher JIF than the three main journals in astronomy [10]. However, those journals provide fewer good data retrieval services [Int-Journal]. Moreover, Int-Faculty12 reports

that in such high-impact journals, often high-quality papers are turned down in favour of papers on sexy topics, which may even contain mistakes.

> "They take these [flashy] results but not the good things. [...] So that's the obsession with journal impact factors. We all know how little journal impact factors mean." [Int-Faculty2]

The peer review process is meant to find mistakes and help improve the quality of the paper, but as Int-Faculty9 explains, reviewers are drowned in other tasks, such as publishing themselves and do not have time to "do the work for the authors". Several interviewees (e.g., Int-PhD1, Int-PhD2, Int-PhD3, Int-Postdoc2, Int-Postdoc3, Int-Faculty2 and Int-Faculty7) report that the decrease in research quality is a result of the publication pressure:

> "There are works that are very important and are published very quickly, and perhaps they are not quality works. There is a lot of publishing pressure. And sometimes the quality goes down because we have to publish like maniacs and because, you know, there is no written standard. But we all know that there must be standards.", "This person has to publish quickly because maybe he or she needs to get more papers out to get a position. So, the quality goes down." [Int-Postdoc2]

As pointed out by Heuritsch [10], there are no external incentives to publish reduction code or data and due to the publication pressure, the intrinsic motivation to do so is often not sufficient. Zuiderwijk and Spiers [35] (p. 233) write that "in the academic system publications are valued more than datasets [...]. This may lead to lower quality research". Int-PhD2 confirms that, in her opinion, the publication system does not encourage good quality research, because:

> "No journal requires these things like they don't require your codes, they don't require that you actually give them step by step what you are doing. So I think as long as that doesn't change as well so no change."

Int-PhD3 explains that it is difficult to know whether a paper is based on good quality data or an error-free code, when their publication is not encouraged. As a reader you do not know how well checked-through all the material that went into the paper is and it is difficult to replicate the analysis [e.g., Int-PhD1, Int-PhD2, Int-PhD3, Int-Faculty1 and Int-Faculty2]. Errors may propagate (Int-PhD3; cf. [2]), and so when one cannot check the analysis step-by-step, one cannot judge the quality of the analysis. The main reason for low replicability is, again, the publication pressure. Int-Faculty1 guesses that 50% of the papers are not replicable, while replicability is an important criterion for him as a referee. Int-PhD2 explains that papers generally are not transparently written and even when one writes an email to the authors for clarification, they often cannot answer the questions sufficiently anymore to be able to replicate the study: "So you cannot trust any, even the published papers apparently." At the same time, [36] (p. 57) refer to the "significant numbers of cases in which negative or inconclusive results do not turn into publications" and regard those as "symptoms of workload pressure in the community". The authors conclude: "This reflects what may be a growing cultural problem in the community as scientists tend to concentrate on appealing results, especially if they have limited resources, and need to focus predominantly on projects that promise to increase their visibility".

The publish-or-perish imperative, does not only lead to lower quality papers, but also to an information overload [26]. Int-Postdoc2 explains that, due to the quantity of papers nowadays, it is difficult to sift through them and decide which ones to use.

> "The only thing I guess the one negative aspect is the pressure on people to publish. And-and I think it's not very healthy because we're getting a lot more lower quality papers and it becomes harder because you have to read all those papers to find relevant information. So probably the pressure on publishing is not a good thing." [Int-Postdoc3]

Herrmann [37] writes that the total publication rate showed a particularly steep increase after WW2, but it exhibited a "steady growth tendency" throughout the 20th century. Publication pressure is only one of the reasons for increased productivity. Another could be that papers are easier to write and publish than in the past, given that there is more available data and that there are more collaborations [11]. Increased availability of data, however, also decreases replicability, since astronomers have less time to replicate each other's results the more papers and data are available [Int-Faculty8]. Davoust and Schmadel [11] mention a third reason for the increased total publishing activity of astronomers, which is the increasing absolute number of astronomers. According to the authors, an individual's relative publication rate may increase with age due to increased competence, more co-authors, and the Matthew Effect.

### 3.2.2. Biased Funding Decisions

The fact that the Matthew Effect and luck play a big role in acquiring capital [30] is also a reason why astronomers often perceive funding decisions as unfair, biased and not very objective (e.g., Int-PhD3, Int-Postdoc1 and Int-Postdoc2). For that reason, Int-Postdoc1 explains that one has to learn not to take funding decisions personally. Int-Faculty9 understands why some would find funding decisions unfair; however, she points out that the same people who may complain about them, may also serve on a panel and make the exact same decisions. Because allocation of funding also needs to be based on how important the research is for the taxpayer, and whether it fits with strategic considerations with regards to the decadal survey or other national interests, the process may be perceived as biased. Furthermore, it is not easy to receive funding when one wants switch subfield, which may lead to overspecialisation.

> "I think it leads to a lot of stove-piping in the field [...]. To stay competitive, [what] all people can do is to continue to dig down deeper in their hole, in their discipline. Like they have to keep proposing for the thing they're most expert on, instead of branching out into other areas. Like branching out has to happen on sort of other you know, side side projects or other funding. It's very, it can be really difficult to get enough of a name for yourself in that area [...] That's not a good thing. That's a- Yeah, and that's a disadvantage of the field like that. I don't think at this point it's pretty diversified." [Int-Faculty12]

Performance indicators define what counts as good research, and what is not measured is not perceived as worth someone's time. For example, fewer astronomers are willing to do something for the common good if it is not directly measured by some metric [Int-Faculty6]. At the same time, collaborations and networking has become increasingly important during past decades, because they serve as trade markets for all kinds of capital.

### 3.2.3. Stress

Not surprisingly, the pressure to publish and the need to comply with institutional norms, may lead to stress. Many interviewees [e.g., Int-PhD2, Int-Postdoc2, Int-Faculty7 and Int-Faculty8] report that job uncertainty causes them emotional stress (cf. [38]). Int-Faculty8 even suffered from depression, because he felt no permanent position was in sight and Int-Faculty7 describes being rejected for postdoc positions as the "most difficult phase" in his astronomy career. Int-PhD2 claims that this uncertainty is the biggest problem for her in academia—even bigger than publication pressure. Like Int-Postdoc2, she would like to reach tenure, but is not sure whether she can accept moving to a different country for every postdoc position. Int-Faculty9 reports that the required flexibility to move and working on weekends can be difficult on relationships:

> "But it does seem like I do have to put in extra time as a lot of the scientists do. To get done what we need to and to stay competitive. [...] And it could be difficult on relationships because I know for my husband he's used to just working a

normal workday, and so—Whereas for scientists it's not just a job, it's kind of a lifestyle."

She is aware that she could have advanced her career faster without having kids, but she is happy about the choice she made and accepted that sometimes one has to decide between a happy family life and a good publication-/citation rate. Int-Faculty8 describes having "heard some horrible stories of how people just make terrible sacrifices" to get tenure:

"I mean, I've had several friends of mine going to the States and, yeah, then go for tenure, but not succeeding. And it's just, doesn't sound like a very pleasant to them, I'll say so."

At the beginning of her PhD, Int-PhD2 feared losing priority, but now she is aware that her topic isn't "sexy" enough for that fear to be justified. If she were working on a sexy topic, she would feel the pressure from the competition, however. Int-Postdoc2 explains that she feels that constant fear of not being good enough, resulting from competition for resources, even though she knows that their allocation also depend on luck. Int-PhD1 reports that he does not "feel comfortable in this system". Int-Faculty5 explains out of personal experience that for early career researchers it is difficult not to take negative referee reports personally, because negative feedback often is perceived as failure and that is connected to fear [Int-Faculty11] because every publication counts.

"So, you really didn't like what you heard and you feel personally attacked, you know. And it's kind of it takes a while to get over that. And it certainly as a young researcher, when you see your first referee report of that type, it can be extraordinarily dispiriting and you need to have a good supervisor who says 'Don't worry, this is quite common. Referees will often attack weak points, but that's part of the scientific process.' [. . . ] And this could have quite a negative effect on people who are trying to start off in the field and who go like, 'Well, he hates me. I'm useless.'" [Int-Faculty8]

While established researchers [e.g., Int-Faculty2] have the opinion that early career researchers shall make their own mistakes to grow and carve out their own niche, this sort of freedom is hardly supported by the publish-or-perish culture. Int-Faculty10 reports that it is "a waste of resources" that the publication pressure cuts the ground under young researcher's free minds. According to Ruocco et al. ([39]: p. 2), "innovative ideas are the results of a random walk in our brain network" and stress may supress that kind of creativity. As a result, the pursuit of risky and innovative ideas is not encouraged, even though that is the science needed by society [40].

### 3.2.4. The Evaluation Gap in Astronomy

When looking at the effects that indicator use in evaluation procedures has on research behaviour and quality in astronomy, I conclude that the initial discrepancy between what astronomers define as good research quality, as opposed to what is measured by indicators, which is present in the astronomer's *situation*, produces an *evaluation gap* at the institutional level. Through constituting what counts, indicators shape individual actions resulting in the *collective phenomenon* of an evaluation gap. That is how indicator use indeed has *constitutive effects* [9] on the knowledge production process in astronomy. At the same time, the evaluation gap remains present since the need to serve indicators only alters the priority of the secondary code with respect to the primary code and not the content of the primary code itself (cf. [2]). The *transformation rules* that lead to the evaluation gap hence comprise the need to perform a balancing act between primary code versus secondary code (Section 3.1.2). As I pointed out in Section 3.1, when most actors agree with the norms at play in an institution, transformation rules simply assume a compliant behaviour of the actors. However, as I have shown, in astronomy behaviour is *conform* and *innovative* at the same time. Astronomers use gaming strategies (Section "The Indicator Game"); To remain competitive, institutionalised means are used in *innovative ways*, such as salami slicing or

going for easy publications, which allows them to prove their performance on paper. In conclusion, the following picture arises (Figure 2): The intention of decision makers to use performance indicators is for researchers to be accountable and produce good quality research ("Good quality research and performance" in Figure 2). Thereby decision makers assume simple transformation rules, where the institutionalised performance indicators lead to compliant behaviour. However, indicators quantify quality [3], and thereby transform qualities into capital that counts. Capital can be targeted, by gaming strategies. Hence, gaming targets the measure. Goodhart's law—"when a measure becomes a target, it ceases to be a good measure"—then is the *first transformation rule* I observe. Further, the discrepancy between the astronomers' intrinsic and extrinsic motivation makes them perform a balancing act, where primary and secondary code are in competition with one another. Astronomers solve this tension relationship (i.e., cognitive dissonance) by rationalising the pursuit of the secondary code at the short-term cost of the primary code, in order to focus on the primary code in the long-term. The art of the balancing act is to choose how much of the value "primary code" can be compromised in favour of the value "secondary code". The balancing act is the *second transformation rule*. Finally, the *third transformation rule* results from compromising the primary code, which corresponds to a compromise of research quality for the sake of quantity. The resulting *collective phenomenon* brings about information overload and sacrificed research quality. On a phenomenological level, I hence observe constitutive effects on what counts as good research (see Figure 2) and an evaluation gap between the intended ideal and the resulting collective phenomenon.

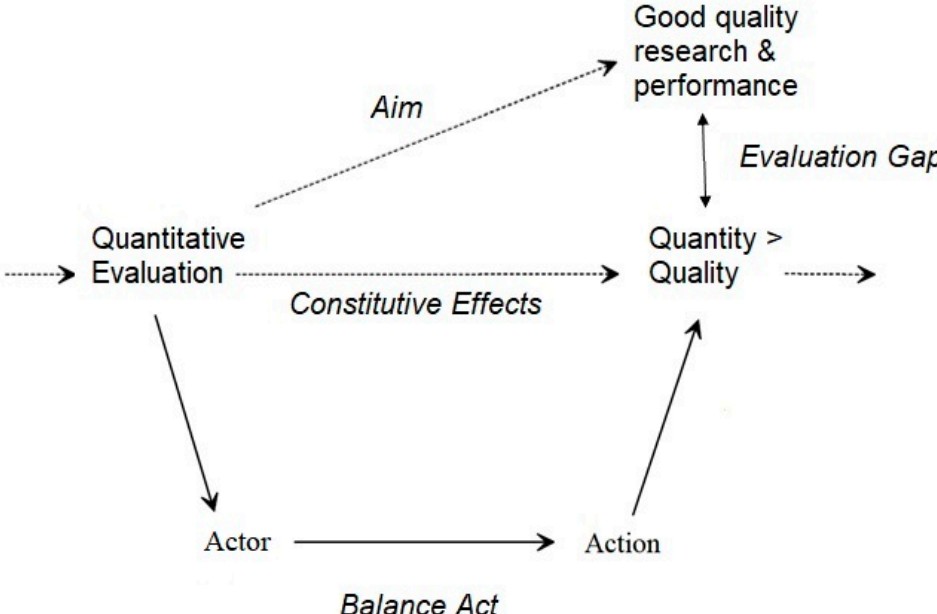

**Figure 2.** The evaluation gap in astronomy as the resulting collective phenomenon based on a predominant evaluation through performance indicators. This illustration is created by the author and based on Coleman's boat [41].

As a summary, the evaluation gap is characterised through avoiding risky topics in favour of sexy ones; i.e., favouring safe publications over potentially innovative outcomes. That is despite astronomers pointing at the fact that in science outcomes cannot be predicted (cf. [42]). Further, due to publication pressure and a lack of external incentives to publish data and reduction code in a transparent way, replicability of papers is sacrificed. Immature publications and salami slicing may further add to a decrease in research quality.

"If I can predict the outcome, it's not science. And several times I've made the right choice to take some risk and just to explore. That's how you find new things." [Int-Faculty2]

"I basically want to do research where people can base-. There's this saying like, 'I was standing on the shoulders of giants.' Basically that's what I want. I hope research would be that you could easily build on previous works. Where you can-. At the moment people just trying to do the same stuff because other people haven't made available their way of methods or codes, and they have to make it up all over again. And then you're only incrementally enhancing the field, and you should really be- [...] And to make something even bigger which later on, in ten years becomes yet again, becomes trivial, and you can build something more. I'm basically seeing it as standing on the shoulders and making a pyramid of knowledge." [Int-PhD3]

"But then there's the frustration of well, you know I'm writing this journal article [...] not just to tell a story or to, to improve [knowledge]. But I'm also doing it to advance my career. And so that ends up being like a [bottleneck] on that sharing, right? It ends up being, you know, 'I don't want to share this material because I have to then write another paper on it'. And which might have some legitimacy or might have more if you said something like, have a student who needs to finish the PhD. And so they need to [...] to safely finish their project. [...] And so then the journals become, feel like-, or at least feel like—I don't know if the journals do—but I feel like that the prioritization is about helping people to publish papers rather than having people to participate in this, this networking of resources. And so, when we talk about that inconsistent loop, you know it might be useful if people got credit for the releasing of data, rather than for the writing of papers." [Int-Journal]

"It's not valid to assume that the scientific community wouldn't want to serve the societal benefits. Think about why people are doing it. If researchers in universities were in for the money,—come on, they would better do something else, it's definitely not the best way to become rich. So they want to achieve something,—why do they want to achieve this? Don't they want to make some difference in this world? So isn't that a societal benefit? They want to make the whole process as efficient as possible. The general interests of a lot of researchers are much aligned to the genuine public interests. Why then do we have a system that is completely offset on this? It not only doesn't make sense, it's also very inefficient. You know, we're publishing more and more and more, the quality doesn't get better, and the quality standards drop in my opinion. We have an inflationary system because we attach larger and larger numbers of research excellence to it, it doesn't mean anything.—It's becoming worse and worse year by year." [Int-Faculty2]

## 4. Discussion

I adopted the *expected utility theory* as an *action theory* to explain how, both intrinsic and extrinsic motivational factors, resulting from the four components of the situation, play together to explain astronomer's research behaviour. I have shown that this can be best described as a balancing act between pursuing the primary and secondary code of science, whereby gaming strategies are employed in order to meet the requirements of the evaluation system. The initial discrepancy between the intrinsic values of an astronomer and what counts as good science as measured by a performance indicator then constitutes, by means of the *logic of aggregation*, an evaluation gap on the institutional level.

To survive in academia, astronomers feel they do not have a choice other than complying to the institutional norms, such as being evaluated by performance indicators; they accept it as part of "the system" [2]. The balancing act then is the art of playing the indi-

cator game (to serve the secondary code), while at the same time compromising research quality as little as possible. Gaming strategies shall give the appearance of *compliance*, while institutionalised means how to achieve a good bibliometric record are used in *innovative ways*, such as salami slicing or going for easy publications. They best fit the third way of how to resolve a cognitive dissonance: by favouring the pursuit of the secondary code in the short-term, one hopes to get into a position where one can focus on the primary code in the long-term. This leads to astronomers reporting an overall decrease in research quality.

Before reaching the final conclusions, I would like to come back to the concept of a *moral economy*. Atkinson-Grosjean and Fairley [14] argue that RCT cannot fully explain the moral economy of astronomy. I disagree. The authors draw that conclusion by making a wrong interpretation of their observations; "rational choice theory undermines many of the values and ideals thought constitutive of the practice of science" (ibid.: 162). This is wrong, because RCT cannot undermine values. Just like "the adjective 'moral' is descriptive not evaluative" (ibid.: 148) in the concept of a moral economy, RCT does not define which action has value and which does not. Assuming that, according to RCT, an act is more valuable when it is rational in our daily use of the word (i.e., high investment of cognitive resources), is a severe misunderstanding of the theory. Instead, RCT advocates for an efficient use of resources. In fact, RCT explains, why in most cases it is actually very rational, not to invest cognitive resources and follow scripts instead. Decreasing our cognitive load by re-parametrizing the situation is one of the very functions of institutional norms (cf. [20]). Hence, RCT is a tool to explain and not to prescribe how an actor acts, by reconstructing the values the person holds and what external conditions are present. This is why I claim that, by employing RCT, one can explain the moral economy of astronomy. Kohler [17] points out that studying "scientific practices requires attention to material culture (instruments, research tools, and methods) and moral economy (social rules and customs that regulate the community), and how these two dimensions work together in a particular line of work" [14] (p.151). By applying RCT I studied both, the material culture and the moral economy of astronomy. McCray [13] (p. 685) employs the concept of a moral economy—"the unwritten expectations and traditions that regulate and structure a community"—"as an analytical model to examine how astronomers and science managers allocate resources." This is also what I have done. In fact, I have shown how indicator use shapes the moral economy in astronomy. I agree with McCray [13] (p. 688), that the "objects of value" of astronomy's moral economy include "an adequate amount of observing time, resources to build and operate new telescope facilities, and funding for one's research". That is how astronomical instruments "shape research opportunities, and they affect careers and institutions" [15] (p. 26) and hence they have constitutive effects on how science is performed. I further agree with [13] (p. 702) that "allocation of resources is not a democratic process"—it is based on political and strategic considerations, as well as performance indicators. I did observe that people from institutes who were involved in building a telescope or who come from prestigious universities have privileged access to telescopes, but did not test whether there is a historical distinction between the "Haves and Have-Nots" like McCray [13] and Atkinson-Grosjean and Fairley [14] did. Furthermore, McCray [13] (p. 686) writes that "the moral economy of astronomy functions through negotiations and compromises". I found the balancing act as an important example for a way to compromise between intrinsic and extrinsic motivation to publish papers.

The aim of this paper was to provide an integrated theory about what effects indicator use has on research quality in astronomy, by means of employing RCT. I conclude that the current structural conditions in astronomy (as outlined by Heuritsch [10]), incentivise astronomers to focus on quantity rather than quality with regards to their publications. To understand the mechanisms that I have outlined in this paper is important for policy makers and funders if it is their aim to provide conditions that encourage quality research instead. Although employing RCT, this is a solely qualitative study, which is not statistically representative and makes inference to causality rather difficult. Future studies may take advantage of RCT's full potential and quantify and verify this study's findings. Moreover, I

suggest to conduct an evaluative inquiry [4], where astronomers are included in development of alternative indicators and incentive structures that make for a research environment which encourages quality science. This process will have to face the challenge that any performance indicator quantifies quality and hence is falls an easy prey to Goodhart's law. However, I am confident that by having analysed the structural conditions and mechanisms at play in astronomy today, this is an important contribution to the development of new ways of resource allocation that encourage quality research over quantity.

**Funding:** This study was performed in the framework of the junior research group "Reflexive Metrics", which is funded by the BMBF (German Bundesministerium für Bildung und Forschung; project number: 01PQ17002).

**Data Availability Statement:** Anonymized Interview Data is available upon request.

**Acknowledgments:** First: I would like to extend my gratitude to the 19 interviewed astronomers for their time and openness. Second, Thea Gronemeier and Florian Beng—the students assistants of the junior research group "Reflexive Metrics" I am part of—were a huge help all along the way; not only by assisting me with the rather monotonous research tasks, but also being a great mental support. Finally, a big thank you for the people who hosted me during the first COVID-19 lockdown, during which this paper was written. It was an incredibly productive time full of professional and personal growth.

**Conflicts of Interest:** The author declares no conflict of interest.

## Appendix A

**Table A1.** Interview Sample Structure and its Represantivity. The labels reveal whether the interviewee holds a PhD, Postdoc, Faculty or Journal position. The division between Global Nort and Global South country of employment was made on basis of the following classification list, accessed on 10 September 2021: https://meta.wikimedia.org/wiki/List_of_countries_by_regional_classification). Comparables [5] represent those astronomers who have an IAU membership. IAU, the International Astronomical Union, is the biggest professional astronomical society with a total of 13,574 members (as of 15 September 2020). From the comparables, I can infer that this sample obtains a good representability, with female astronomers slightly overrepresented. Only >60 year old astronomers are underrepresented. However, this may be due to the fact, that astronomers may remain (active) members of the IAU even after retirement. The list is sorted in chronological order with respect to the date of the interview.

| Label | Gender | | Age | | | Country of Employment | | Date of Interview |
|---|---|---|---|---|---|---|---|---|
| | Female | Male | <35 | 35–60 | >60 | Global North | Global South | |
| Int-Faculty1 | 0 | 1 | 0 | 1 | 0 | 1 | 0 | 27 June 2017 |
| Int-PhD1 | 0 | 1 | 1 | 0 | 0 | 1 | 0 | 27 June 2017 |
| Int-PhD2 | 1 | 0 | 1 | 0 | 0 | 1 | 0 | 28 June 2017 |
| Int-Faculty2 | 0 | 1 | 0 | 1 | 0 | 1 | 0 | 28 June 2017 |
| Int-PhD3 | 0 | 1 | 1 | 0 | 0 | 1 | 0 | 29 June 2017 |
| Int-Journal | 0 | 1 | 0 | 1 | 0 | 1 | 0 | 27 August 2018 |
| Int-Faculty3 | 0 | 1 | 0 | 0 | 1 | 1 | 0 | 27 August 2018 |
| Int-Faculty4 | 0 | 1 | 0 | 0 | 1 | 1 | 0 | 28 August 2018 |
| Int-Faculty5 | 1 | 0 | 0 | 1 | 0 | 0 | 1 | 29 August 2018 |
| Int-Faculty6 | 0 | 1 | 0 | 0 | 1 | 1 | 0 | 29 August 2018 |
| Int-Faculty7 | 0 | 1 | 0 | 1 | 0 | 0 | 1 | 29 August 2018 |
| Int-Faculty8 | 0 | 1 | 0 | 1 | 0 | 1 | 0 | 2 October 2018 |

**Table A1.** *Cont.*

| Label | Gender | | Age | | | Country of Employment | | Date of Interview |
|-------|--------|------|-----|-------|-----|-------------------|--------------|-------------------|
| | **Female** | **Male** | **<35** | **35–60** | **>60** | **Global North** | **Global South** | |
| Int-Faculty9 | 1 | 0 | 0 | 1 | 0 | 1 | 0 | 26 October 2018 |
| Int-Postdoc1 | 1 | 0 | 0 | 1 | 0 | 1 | 0 | 9 November 2018 |
| Int-Faculty10 | 0 | 1 | 0 | 0 | 1 | 1 | 0 | 9 November 2018 |
| Int-Faculty11 | 0 | 1 | 0 | 1 | 0 | 1 | 0 | 21 November 2018 |
| Int-Postdoc2 | 1 | 0 | 0 | 1 | 0 | 1 | 0 | 5 December 2018 |
| Int-Faculty12 | 0 | 1 | 0 | 1 | 0 | 1 | 0 | 7 December 2018 |
| Int-Postdoc3 | 0 | 1 | 0 | 1 | 0 | 0 | 1 | 7 December 2018 |
| **Sample total** | **5** | **14** | **3** | **12** | **4** | **16** | **3** | |
| **Sample in percent** | **26%** | **74%** | **16%** | **63%** | **21%** | **84%** | **16%** | |
| **Field [6] in percent** | **18%** | **82%** | **4%** | **57%** | **39%** | **80%** | **20%** | |

## Appendix B

**Table A2.** Codes used in the analysis. ">" denotes a sub-category of the previous code.

| Code | Memo |
|------|------|
| Authorship | Mentioning of the relevance of authorship (includes also order of authorship) |
| CAREER Clarity/Expectations | Mentions of how aware one was of the expectations that come with one's aspired career in academia and how consciously one chose a career in academia |
| Collaboration | Mentioning of the importance and practice of collaborations |
| Competition | Mentioning of the importance and practice of competition |
| Curiosity | When curiosity is mentioned as a reason for doing research (e.g., "Wanting to understand") |
| >Puzzle | Curiosity may express itself through a variety of other forms. One of which is the love for "Puzzle solving" [43]<br>When a reason for doing research involves love for puzzle solving (e.g., "intellectual challenge") |
| Epistemic Subculture | Mentioning of relevance of the topic of research, and whether one works rather in Instrumentation/Observation/Theory |
| Failure | Mentioning of the meaning of failure (e.g., to publish, to obtain research results, to take the next career step) to the interviewee |
| Funding | Mentioning of the importance/practice of receiving funding |
| Gaming | Mentioning of any kind of gaming strategy, targeting, capacity of convincing funders/politicians to invest in one's research (e.g., "Sales men"), salami slicing, etc |
| Impact | Mentioning of the relevance/meaning of one's research to have "impact" |
| Indicator | Mention of all sorts of indicators (publication rates, H-index etc) |
| >Awareness | How much do astronomers themselves reflect on being measured? |
| Integrity | Mentioning of encouragement or threat to research integrity. I.e. also use for opposite like: Fraud, Fake, Cheat |
| Limitations/Obstacles | Mentioning of all sorts of things that may be restricting for conducting research (e.g., time, resources, staff, etc) |
| Luck | Mentioning of the role of luck as opposed to merit in receiving funding/positions etc |

**Table A2.** *Cont.*

| Code | Memo |
|---|---|
| Matthew effect | Mentioning of the Matthew Effect phenomenon; Incidences where resources (e.g., funding) were more dependent on prestige/quantitative output than the quality of the research; Situations where past output determines future success; Also mentioned as "the chicken or egg problem" by an interviewee in Heuritsch [2]: funding is difficult without having acquired acknowledgement for previous research and acquiring such acknowledgement is difficult without funding in order to perform research. |
| Negative results | Mentioning of the relevance/publishability of negative results, incl. non-detections |
| Output orientation | Displaying need to pay attention to output. E.g. when certain output is mentioned as a basis of one's assessment. Including perceived expectations on one's output. |
| >Citation rates | Mentioning of the relevance of citation rate |
| >Publication rates | Mentioning of the relevance of publication rate |
| Politics | Mentioning of the importance of political aspects of hiring/how the research is organised |
| Pressure | Perceiving any kind of pressure (e.g., to publish/to receive funding) |
| Prestige | Mentioning of the relevance of prestige/being known in the community |
| Publication | Mentioning of the importance/practice of publishing |
| Purple | Interviewee expressions that are particularly demonstrative/well expressed so that they can be used well for citing in the resulting paper. |
| Quality | Mentioning of the relevance of research quality and effects on research quality |
| >3 criteria | If at least one of the 3 criteria that Heuritsch [2] identified as the astronomer's most important quality criteria is mentioned |
| Replicability | Mentioning of relevance of replicability and also practice of ensuring replicability |
| Riskiness | Mentioning of the relevance and practice of taking risks in research |
| Sexy topics | Mentioning of relevance of so-called sexy/fashionable/flashy topics |
| Uncertainty (career) | Expressions of uncertainty and associated (negative) feelings about one's career in academia |
| Temporality | When time(ing) is important |

## Appendix C

**Table A3.** Interview Questions.

| Concept | Research Theme | Question # | Interview Question |
|---|---|---|---|
| **Introduction** | Background | 1 | Please tell me about your background—from how you decided to go into Astronomy to how you got into your current position. |
| **Introduction** | Astronomy | 2 | What is special about astronomy for you? |
| **Evaluation** | Necessary conditions to perform science | 3 | What do you need to do to be able to do your research? >What processes are involved in your research/What resources do you need?/What obstacles do you face in the |
| **Evaluation** | | 3.10 | process? |
| **Data** | | 4 | Do you search for published data (do you use data of others?) |
| **Data** | Challenges of the data | 5 | What are the challenges in getting the data you need? |
| **Data** | acquisition process | 5.10 | >Telescope |
| **Data** | | 5.20 | >Database |
| **Data** | | 7.00 | Do you usually publish data with your research results? |
| **Data** | | 7.10 | >Do you publish data separately? |
| **Data** | | 7.20 | >Do you think this happens often in your discipline? |

**Table A3.** *Cont.*

| Concept | Research Theme | Question # | Interview Question |
|---|---|---|---|
| **Evaluation** | Practices: Publishing | 8.00 | What are the criteria to publish your work? |
| **Evaluation** | | 8.10 | >Can you tell me a few examples of your successes and disappointments regarding publishing? |
| **Evaluation** | Practices: Funding | 9.00 | What are the criteria to receive funding? |
| **Evaluation** | Practices: Funding | 10.00 | Do you find funding decisions fair/reasonable? |
| **Evaluation** | Practices: Funding | 11.00 | Can you tell me a few examples of your successes and disappointments regarding acquiring funding? |
| **Evaluation** | Other experiences with evaluation | 12.00 | You told me about about postition, getting money etc –are there any aspects of your scientific work where assessment plays a role? |
| **Evaluation** | | 12.10 | >In what way do you face situations in which your work or parts of what you do is appraised directly or indirectly by others? (e.g., Yearly appraisals with your supervisor/Peer review for funding applications/Mid-term reviews for projects) |
| **Evaluation** | | 12.20 | >What role do indicators play in assessments? |
| **Evaluation** | Temporality | 13.00 | If you were starting your career now how would things be different for you? |
| **Notions of quality** | Introduction to this block | 14.00 | Well, thank you for telling me about your research interests and activities. In this last section of the interview I'd like to ask a few questions about ideas of 'quality' in scientific work. Sometimes people talk about 'quality' in research or in publications, and I'd be interested to know how you think about it. Perhaps I could start with the question—what, in your view, makes a good scientific paper |
| **Notions of quality** | | 15.00 | Would you say 'quality' is something you judge when reading a paper? |
| | What is research quality for the astronomer? | 15.10 | >What makes a good paper in your opinion? |
| **Notions of quality** | | 15.20 | >When you read a paper, how do you decide if it is a good article? |
| **Notions of quality** | | 16.00 | Could you give me an example of a "high quality" work? |
| **Notions of quality** | What is quality for "the system"? | 16.10 | >How was this work perceived by your peers/the community? |
| **Notions of quality** | | 17.00 | What is your highest quality paper? |
| **Notions of quality** | | 17.10 | >Do you think it was the most important one? |
| **Notions of quality** | Divergent notions of quality? | 17.11 | >>Was it the most influential one? |
| **Notions of quality** | | 17.20 | >Does it have most citations? |
| **Notions of quality** | | 17.30 | >(If no:): how would you describe the difference (e.g., between a paper that is good and a paper that has the most citations)? |
| **Evaluation inquiry** | Ideas for alternative evaluation/indicators? | 18.00 | What would you wish that would be measured? |
| **Notions of quality** | What is quality for "the system"? | 19.00 | What makes a good application/funding proposal? |
| **Evaluation inquiry** | Ideas for alternative evaluation/indicators? | 18.00 | What would you wish that would be measured? |

## Notes

[1] Note here that Bourdieu's [22] concept of capital and Esser's concept of "control of resources" refer essentially the same phenomenon: An actor has or has not access to various forms or capital (resources), such as money, time, social network and has incorporated certain knowledge and values.

[2] "I mean so that's how you do good science you just look at your data. [...] What the data are telling you. " [Int-Faculty2]

3 "Yeah so we're back to the thing. Is this the true academic quality? I mean the true academy hasn't changed. Quality is quality. Or what people would call quality. Or people, mistake quantity for quality. Any metric measures quantity, it doesn't measure quality." [Int-Faculty2]

4 https://blogs.lse.ac.uk/impactofsocialsciences/2018/11/29/the-evaluative-inquiry-a-new-approach-to-research-evaluation/ (accessed on 17 May 2023).

5 https://www.iau.org/administration/membership/individual/distribution/ (accessed on 17 May 2023).

6 https://www.iau.org/administration/membership/individual/distribution/ (accessed on 17 May 2023).

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
