# Peer review of "The Evaluation Gap in Astronomy—Explained through a Rational Choice Framework"

_publications, doi:10.3390/publications11020033_

Round 1

Reviewer 1 Report

This article is coherent, actual, and relevant not only for the scientific area of astronomy but also for research on the Evaluation Gap theme.

Research questions are clearly identified. The theoretical foundation is updated and aligned with the objectives of the study.

The selected methodology is adequate and allows for reaching relevant results. The collected data is sufficient, and the codification process is explained.

The discussion of the results is articulated with the theoretical foundation.

The argument is solid and the article raises current questions about the behavior of scientists and about the impact of evaluation on the necessary match between intrinsic and extrinsic motivations.

the use of this framework can be replicated with scientists from other scientific fields and can help in the management of research in research centres, both in Universities and in companies.

I reinforce that it was an article that gave me pleasure to read, review and reflect on its content, which allowed me to learn.

Author Response

Dear Reviewer 1,

Thank you for your supportive & positive feedback and your time!

Best wishes,
Julia

Reviewer 2 Report

The study is an important contribution to the discussion about how and why scientific researchers cope with the tension between following institutional demands for academic productivity and efforts to do quality research. The author opts for the rational choice framework and generates an argument in favor of this explanatory model while supporting it with empirical evidence based on interviews with astronomers. I think it is an honest and useful intellectual work that deserves to be published.

Author Response

Dear Reviewer 2,

Thank you for your supportive & positive feedback and your time!

Best wishes,
Julia

Reviewer 3 Report

The main question addressed by the research is how observational astronomers manage the balancing act between intrinsic values and the requirements of evaluation procedures, and how this affects research quality in astronomy. Specifically, the study analyzes the existence of evaluation gaps and their consequences for the relationship between intrinsic and extrinsic motivations to perform research.

The topic is both original and relevant in the field. The study addresses the concept of evaluation gaps, which capture potential discrepancies between what researchers value about their research and what metrics measure. The existence of evaluation gaps and their consequences for research quality are important topics in academia, and this study specifically focuses on the case of observational astronomers.

Furthermore, the study fills a specific gap in the field by analyzing the evaluation gap from a rational choice point of view and shedding light on the workings of the balancing act and its consequences on research quality in astronomy. The study also provides insights into the strategies used by researchers to comply with institutional norms and achieve good bibliometric records while compromising research quality as little as possible.

The conclusions seem to be consistent with the evidence and arguments presented, and they address the main question posed in the study. However, a full evaluation of the study's conclusions would require a thorough analysis of the methodology, data, and results presented.

Some general suggestions that the authors could consider to strengthen the methodology are:

Increase the sample size: Although the study conducted semi-structured interviews with international astronomers, increasing the sample size could improve the generalizability of the findings.

Increase the diversity of participants: The study could include a more diverse group of participants, such as junior and senior astronomers, astronomers from different geographic locations, and those from different types of institutions.

Use multiple data collection methods: The study could use multiple data collection methods, such as surveys and document analysis, to triangulate the findings and increase the validity of the results.

Consider the role of contextual factors: The study could consider the role of contextual factors, such as the size of the institution and the funding structure, in shaping the evaluation gap and its consequences.

The authors should consider these suggestions, among others, to strengthen the methodology and increase the rigor of the study.

It is also important to note that the findings of the study may be specific to the case of observational astronomers and may not be generalizable to other fields or contexts.

Overall, the study seems to address an important topic and provides insights into the strategies used by researchers to manage the balancing act between intrinsic values and requirements of evaluation procedures. Further research in this area could help to shed more light on the evaluation gap and its consequences on research quality in academia.

Author Response

Dear Reviewer 3,

Thank you for your assessment and the time you spent on it!

You have some comments with regards to strenghtening the methodology, which I will reply to in the following.

Naturally, this study has been concluded, however you may be pleased to hear that your suggestions are largely covered by my follow-up studies, written in the following order:

  1. Heuritsch, J. Reflexive Behaviour (2021): How Publication Pressure Affects Research Quality in Astronomy. Publications2021, 9, 52. https://doi.org/10.3390/publications9040052
  2. Heuritsch J (2023) Reflecting on motivations: How reasons to publish affect research behaviour in astronomy. PLoS ONE 18(4): e0281613. https://doi.org/10.1371/journal.pone.0281613
  3. Heuritsch J (2021) Towards a Democratic University: A call for Reflexive Evaluation and a Participative Culture. Under Peer review at Frontiers; ArXiv:
    https://doi.org/10.48550/arXiv.2112.08963

All three studies are based on a web-based survey among world-wide astronomers across all status groups. 3509 astronomers completed the survey partly, of which 2011 astronomers did so fully. The first two mentioned studies represent a quantitative analysis of the 1. Evaluation gap and 2. the motivational factors as part of the internal constituents of an astronomer’s situation. This includes the consideration of contextual factors with respect to organizational culture. The third study analyses the open questions of the web-based survey and – interweaving these answers with qualitatitive interviews – focusses on how a more reflexive evaluation and diversifying what counts as output could transcend the evaluation gap and foster research quality.

I hope this satisfies your points regarding sample size, diversity of participants, contextual factors and multiple data collection methods.

Thank you again for your feedback and best wishes,

Julia